# On the Specificity and Permanence of Electroencephalography Functional
Connectivity

**DOI:** 10.3390/brainsci11101266

**Published:** 2021-09-24

**Authors:** Yibo Zhang, Ming Li, Hui Shen, Dewen Hu

**Affiliations:** College of Intelligence Science and Technology, National University of Defense Technology, Changsha 410073, China; zhangyibo@nudt.edu.cn (Y.Z.); shenhui@nudt.edu.cn (H.S.); dwhu@nudt.edu.cn (D.H.)

**Keywords:** functional connectivity, permanence, individual specificity, EEG

## Abstract

Functional connectivity, representing a statistical coupling relationship between
different brain regions or electrodes, is an influential concept in clinical medicine and
cognitive neuroscience. Electroencephalography-derived functional connectivity (EEG-FC)
provides relevant characteristic information about individual differences in cognitive
tasks and personality traits. However, it remains unclear whether these
individual-dependent EEG-FCs remain relatively permanent across long-term sessions. This
manuscript utilizes machine learning algorithms to explore the individual specificity and
permanence of resting-state EEG connectivity patterns. We performed six recordings at
different intervals during a six-month period to examine the variation and permanence of
resting-state EEG-FC over a long period. The results indicated that the EEG-FC networks
are quite subject-specific with a high-precision identification accuracy of greater than
90%. Meanwhile, the individual specificity remained stable and only varied slightly after
six months. Furthermore, the specificity is mainly derived from the internal connectivity
of the frontal lobe. Our work demonstrates the existence of specific and permanent EEG-FC
patterns in the brain, providing potential information for biometric applications.

## 1. Introduction

Brain functional connectivity plays an essential role in neuroscience. It reflects the
complex functional dependence and coupling of neural activity between brain regions [1]. Measures of connectivity can be
recognized by a variety of physiological recording techniques, including magnetic resonance
imaging (MRI), near-infrared spectroscopy, and electroencephalography (EEG). As a new method
of individual differences and pathological research, functional network analysis has
attracted increasing attention from the scientific community [2,3]. However, the individual specificity and permanence of human brain networks
have not been thoroughly studied.

Individuals may have unique and characteristic connectivity patterns. Some previous MRI
studies have demonstrated that functional connectivity networks exhibit high variability
among individuals, serving as “fingerprints” of individuals [4]. fMRI provides good anatomical
resolution and endogenous explanations for individual differences in functional brain
networks, but its temporal resolution is limited [5]. Unlike fMRI, EEG is a practical and convenient
approach to explore the temporal changes in functional brain connectivity, non-invasively
recording neuronal activity at the millisecond level [6].

The increasing application of EEG network research has attracted attention regarding
whether EEG functional connectivity (EEG-FC) can be sustained over time and across cognitive
states. Permanence represents the intraindividual stability of brain activities over a
period of time [7]. Previous
studies, such as those on EEG biometrics, only dealt with single-session data sets. This
study design leads to a concern about whether the accuracy of the recognition represented
the uniqueness of the recognition according to the EEG characteristics of the subjects or
the uniqueness of each acquisition session [8]. At present, some studies have begun to focus on the permanence of biometrics
[9,10]. A study found that the permanence of the
core-specific network structure of EEG-FC in five healthy subjects remained stable for
several days [11]. Some studies
have demonstrated that resting-state EEG-FC represents a powerful method for high-precision
biometric identification purposes [6,12].

However, these studies on EEG-FC patterns recorded only two or three sessions over a few
weeks, which resulted in a lack of variation over a long period in the longitudinal study.
Therefore, it is necessary to establish a sufficient number of recordings to firmly
demonstrate whether specific functional connectivity can remain relatively permanent across
time. In this manuscript, we explored the individual specificity and temporal permanence of
resting-state EEG-FC from 15 healthy participants with multiple experimental sessions over
six months.

Based on the previous fMRI and EEG studies, we had two hypotheses: (1) There may be
individual differences in EEG-FC patterns, which could be verified by identification. (2)
The individual-specific EEG-FC may be relatively permanent across disjoint sessions.
Biometric identification was used as a research strategy in this manuscript to verify these
hypotheses. Our exploratory work may be helpful to reveal the individual differences in
neural electrical signal activity.

## 2. Materials and Methods

### 2.1. Participants

A total of fifteen healthy subjects (24 ± 2 years old) participated in this study.
No history of neurological or psychiatric disorders, migraine, diabetes, or tinnitus was
reported. All participants were right-handed and had normal vision (or corrected vision).
All participants submitted written informed consent after receiving a detailed explanation
of the experimental procedure. The studies involving human participants were reviewed and
approved by the Medical Ethics Committee of 921 Hospital.

### 2.2. Experiment Protocol

The experimental environment was quiet. As shown in Figure 1, the subjects were given experimental
precautions and sat in a comfortable chair 1 m away from the monitor. To reduce eye
movement, the participants were required to look at a single black ‘+’ in
the center of a display screen for at least 2 min and close eyes for another 2 min. During
the recording of the 4 min resting-state EEG, the participants needed to keep still, avoid
blinking, and to not think about other things. Recordings were repeated multiple times for
each subject over a period of six months, including Day 1 (Session A), Day 2 (Session B),
Day 7 (Session C), Day 30 (Session D), Day 90 (Session E), and Day 180 (Session F).

### 2.3. Data Acquisition and Pre-Processing

In the process of creating the dataset, we used Nerusen 32-channel wireless EEG
acquisition equipment from the Neuracle company(Changzhou, China). The position of the
electrodes was shown in Figure 2.
According to the 10/20 international standard system, the electrodes were located at
frontal, central, temporal, parietal, and occipital scalp sites [13] (Fp1, Fp2, F7, F3, Fz, F4, F8, FC5, FC1, FC2,
FC6, A1, T7, T8, A2, C3, Cz, C4, CP5, CP1, CP2, CP6, P7, P3, Pz, P4, P8, PO3, PO4, O1, Oz,
and O2). The reference electrodes were REF and GND. The sampling rate was set to 1000 Hz,
and the electrode impedance was adjusted below 10 kΩ.

EEG data were exported to EEGLAB [14] for pre-processing. In the first step, EEG data were filtered with a
bandpass filter of 1–40 Hz, including a notch filter of 50 Hz. Then, eye movements,
eyeblinks, muscle activity, and other obvious artifacts were corrected by applying an
independent-component analysis (ICA) implemented in the EEGLab toolbox. Based on the
artifact-free dataset (Reject ICs), we applied a spatial Laplacian filter using the
current source density (CSD) toolbox to reduce the effects of volume conduction [15].

### 2.4. Connectivity Measures and Machine Learning

Previous studies have suggested that 2 s is sufficient for functional connectivity of
resting-state EEG data [16].
Therefore, the preprocessed EEG data were segmented every two seconds. We performed the
same process described above for each session dataset. Each session dataset included 15
subjects, 4 min of resting-state EEG signals. In total, this resulted in 120 × 15
segments for connectivity calculations. In order to maximize the retention of individual
brain signal characteristics, we chose to retain the neural activity of all frequency
bands for brain network analysis. The frequency band of EEG-FC was selected as
1–40Hz. We used the Hermes Toolbox [17] to calculate the Granger causality (GC) index and
the mutual information (MI) index.

Granger causality, derived from the definition of causality in statistics, was introduced
into neuroscience to describe brain functional connectivity. For two simultaneously
measured time series signals x(t)
and y(t),
x(t)
causes y(t)
if the former contains information that helps predict the future of the latter. Two time
series x(t)
and y(t)
are modeled by BVAR (the bivariate autoregressive model), which including the past samples
from the time series itself and the other time series. The calculation formula of the
Granger causality index is as follows: (1)$$x(n)=\sum_{k=1}^{P}a_{x|x,k}x(n-k)+\sum_{k=1}^{P}a_{x|y,k}y(n-k)+u_{xy}(n)\;y(n)=\sum_{k=1}^{P}a_{y|x,k}x(n-k)+\sum_{k=1}^{P}a_{y|y,k}y(n-k)+u_{yx}(n)$$
(2)$$\begin{array}{c} V_{x|\overset{ˉ}{x},\overset{ˉ}{y}}=var\left(u_{xy}\right)\\ V_{y|\overset{ˉ}{x},\overset{ˉ}{y}}=var\left(u_{yx}\right) \end{array}$$
(3)$$GC_{y\to x}=ln\left(\frac{V_{x|\overset{ˉ}{x}}}{V_{x|\overset{ˉ}{x},\overset{ˉ}{y}}}\right)$$

In Equations (1) and (2), $a_{ij}\;$
are the model parameters, p is
the order of the BVAR model and $u_{i}$
are the residuals associated to the model, var(.)
is the variance over time and $x|\overset{ˉ}{x},\overset{ˉ}{y}$
is the prediction of x(t)
by the past samples of values of x(t)
and y(t).
Granger causality (GC) from y(t)
to x(t)
greater than or equal to zero. In our calculation, the model order
p was
15. The GC net relationship enabled the detection of directed and reciprocal influences
common in brain coupling [18].
The GC net calculated by our recording data could describe the individual effective
connectivity between distinct electrodes in the resting state.

Based on concepts from information theory, mutual information (MI) measures the
interdependence between two variables, which quantifies the amount of information obtained
from about one random variable by other random variables [19]. (4)$$MI_{xy}=\underset{i}{\sum }p\left(x,y\right)\log\frac{p\left(x,y\right)}{p\left(x\right)p\left(y\right)}$$

In Equation (4), p(x,y)
is the joint probability distribution function of x(t)
and y(t),
and p(x)
and p(y)
are the marginal probability distribution functions of x(t)
and y(t),
respectively. The equation represents the cross-mutual information between
x(t)
and y(t).
The MI of two random variables is a measure of the mutual dependence between the two
variables. If the value of MI is zero, the electrical signals of the two channels were
independent.

As shown in Figure 3, after
constructing the brain network matrix of GC and MI, we used the identification procedures
as a strategy evaluating individual specificity and permanence. We calculated that the
number of net features between all electrode pairs of the GC index and MI index at each
segment was 1024 (32 × 32) and 496 (32 × 31/2), respectively. As a reliable
method, machine learning has been frequently used in previous MRI and EEG studies [20,21]. We used a support vector machine with a radial
basis kernel function (SVM-RBF) algorithm to identify 15 participants. The one-against-one
(OAO) multiclassification method was used in this manuscript. OAO designs a classifier
between any two types of samples. When making classification decisions for an unknown
sample, the voting method is adopted, and the category with the most votes in the category
of the unknown sample.

We utilized the multi-classification method of machine learning for individual
identification. By classifying the subject in a single session data set, we can explore
whether the functional connectivity based on EEG could realize high variability among
individuals. High-precision individual identification could indicate the existence of
individual-specific brain network. In addition, the reproducibility and permanence of
specific brain networks could be reflected by the classification accuracy across disjoint
sessions.

## 3. Results

### 3.1. The Functional Connectivity Network between Individuals Is Specific within
Different Sessions

The accuracy of identification in a single session represents the strength of individual
specificity. High identification accuracy corresponds to strong individual functional
connectivity differences. To explore the specificity of functional connectivity index, we
performed identification classification on each session dataset. A total of 9/10 of each
session dataset was taken as the training set, and 1/10 served as the testing set. Through
ten-fold cross-validation, the 15-classification accuracy of each single session was
obtained. Figure 4 shows multiple
experimental sessions (A–F) conducted on the same group of subjects over a period
of 6 months. The time points are Day 1, Day 2, Day 7, Day 30, Day 90, and Day 180. Figure 5a shows the individual
specificity of functional connectivity from each single session experiment. The data of
the same session experiment are divided into the training set and test set, and the
classification accuracy is obtained by ten-fold cross-validation. The 15-classification
accuracy of the GC network on a single session was 0.89 to 0.96, and the MI network was
0.82 to 0.92, respectively. Here, we show that individuals have unique EEG-FC profiles
similar to fingerprints.

### 3.2. Individuals Have Relatively Permanent Functional Connectivity Patterns across
Long-Term Sessions

The accuracy of identification across sessions represents the permanence of individual
specificity. High accuracy corresponds to the similarity of individual functional
connectivity between subsequent follow-up sessions and first sessions. To explore the
permanence of the functional connectivity metric, we tested it across sessions. Session A
collected on Day 1 was used as the training set. Session B collected on Day 2 was used as
the testing set. The Session C–F data at other time points were used as the test
set, in turn. The classification accuracy of cross-time sessions was obtained. Figure 5b shows the permanence of
functional connectivity. The data of the first-day session is used as the training set,
and the data of other session experiments are used as the test set. The GC network
classification accuracy decreased from 0.939 on the second day to 0.770 on the 180th day.
It shows that the individual specificity remained relatively stable and vary slightly in 6
months. The classification accuracy of mutual information decreased from 0.903 on the
second day to 0.627 on the 180th day, which showing a monotonous decrease.

### 3.3. The Difference in Individual Brain Networks Mainly Comes from the Internal
Connections of Frontal Lobe

According to the 10/20 international standard system, the electrodes were located at
frontal, central, temporal, parietal, and occipital scalp sites [13]. EEG signals collected by different regions of
the brain surface reflect completely different information. To explore the contribution of
different brain regions to GC net specificity, we used machine learning to identify the
features that make important contributions to classification. We extracted the
classification scores of each feature vector and applied canonical correlation analysis
(CCA) to find the related functional connectivity. The functional connectivity of each
pair of electrodes was assigned a weight, indicating its contribution to the
identification process. The major connection is defined as the functional connectivity
with the top 10% of feature weights in the identification by SVM. The weight represents
each pair’s contribution to the identification process. Larger weights indicate
more specific connections.

We used BrainNet [22] to draw
the top view, right view, frontal view, and left view of the specific connectivity
distributed on the brain surface. For clear representation, the colored lines indicate
connections within each brain region, and grey lines represent inter-region connections.
As shown in Figure 6, The larger
spheres in the red network indicate a larger number of major connections in the frontal
lobe region. Different colored spheres represent electrodes in different brain regions.
Compared to other colors of the connection, these red lines indicate strong connections
between electrodes in the frontal region of the brain. The difference in individual brain
networks is mainly derived from the internal connections of the frontal lobe. The results
indicated that individuals have relatively permanent specific connectivity patterns mainly
due to the internal connections of the frontal lobe.

## 4. Discussion

The neural activation and connectivity of non-invasive high-resolution spatiotemporal
patterns have greatly improved our understanding of the individual mechanisms involved in
perception, attention, and learning. Measures of functional connectivity of EEG signals are
increasingly being used to study brain function in spontaneous neural activity. In our work,
the brain network based on GC has the potential to reveal the long-term specific patterns of
individuals. Our work shows that the measurement of EEG-FC at the sensor level can be used
for biometric purposes with high recognition accuracy. In addition, the permanence of EEG-FC
as a biological identifier was further demonstrated. Functional and effective connectivity
measures convey important information about the neural network. Although the EEG network is
generally believed to be fragile, it was demonstrated to be suitable for convenient and
efficient biometrics.

Resting-state is a state in which the brain keeps quiet and awake without performing
specific cognitive tasks. It may be the most basic but essential state of the brain [23]. Our results described the
specificity and permanence of resting-state EEG connectivity measures, which could be
considered in exploring inter-individual differences in brain relationships. However, it is
important to note that this study only focused on the resting-state EEG, and caution is
necessary when extending our results to event-related potentials or EEG of natural stimuli.
It still needs to further explore whether specific EEG functional networks could maintain
permanence under various cognitive tasks such as watching videos, playing games, and
listening to music.

EEG signals collected from different areas of the brain surface reflect completely
different neural activity information within the brain. Different brain regions could
reflect different functional states [24]. Previous fMRI studies using data from the Human Connectome Project have
demonstrated that the frontoparietal network emerged as most distinctive in individual
characteristic connectivity patterns [4]. Although EEG does not have high-intensity spatial resolution, we could
describe the correspondence of brain scalp functional areas through the electrodes on the
surface scalp. The functional network consists of neural electrophysiological activity that
can span multiple scalp regions of the brain [25]. We explored the major connections in the network
by the contribution of each pair of electrodes to the recognition process. The frontal lobes
represent higher cognitive functions and are responsible for learning, language,
decision-making, and emotion [24,26]. Neural activity
in the frontal lobe represents the region where conscious thoughts and decisions occur
[27]. This feature may have
resulted in a stable GC network in the flow of neuro-electrical signals to the frontal lobe
network. In the process of investigating the EEG-FC, our conclusion verifies to some extent
that the frontal network corresponding to EEG is also the most distinctive region of
individual specificity in the resting state. This may help to represent individual brain
networks with more significant and fewer EEG networks. In addition, electrical activity in
the resting state may produce significant functional connectivity networks in various
functional areas. EEG-FC patterns within frontal functional areas may be more assertive in
the resting state. In the future study, it still needs to explore whether the permanence of
individual EEG-FC patterns is independent of the various cognitive tasks across time.

## 5. Conclusions

In this study, we have explored the individual specificity and temporal permanence of
EEG-FC with multiple experimental sessions over a relatively long time. Through our
research, we report the following conclusions. The EEG-FC network based on Granger causality
could exist for a long time as an individual unique connectivity pattern. In the
characteristic connectivity pattern, the internal connections of the frontal lobe network
may play a significant role. These results are conducive to revealing the individual
differences in neural electrical signal activity and promote the application of the
resting-state brain network in biometric identification.

## Figures and Tables

**Figure 1 brainsci-11-01266-f001:**
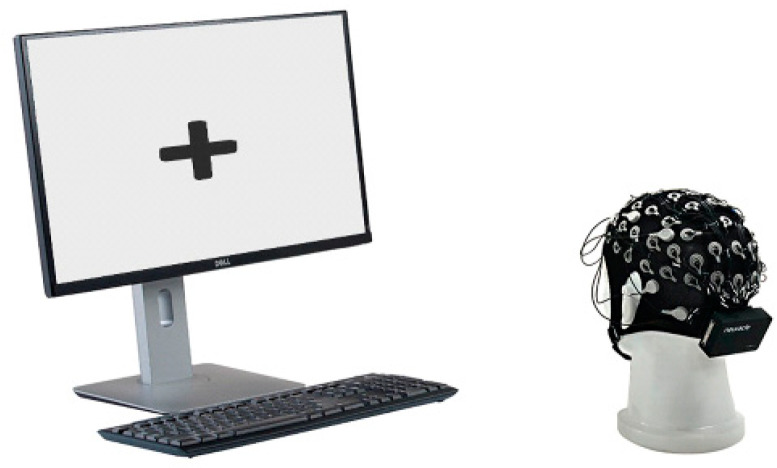
EEG Acquisition.

**Figure 2 brainsci-11-01266-f002:**
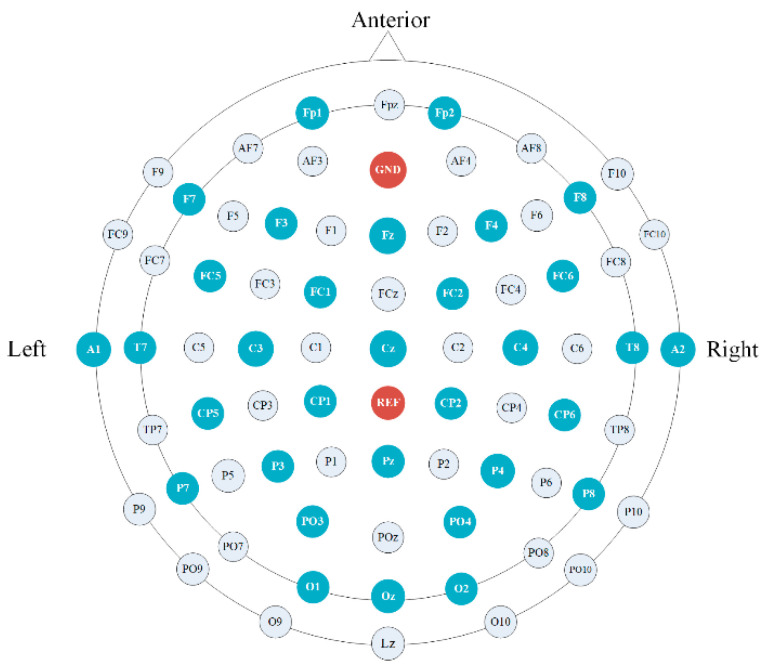
Electrode Placement.

**Figure 3 brainsci-11-01266-f003:**
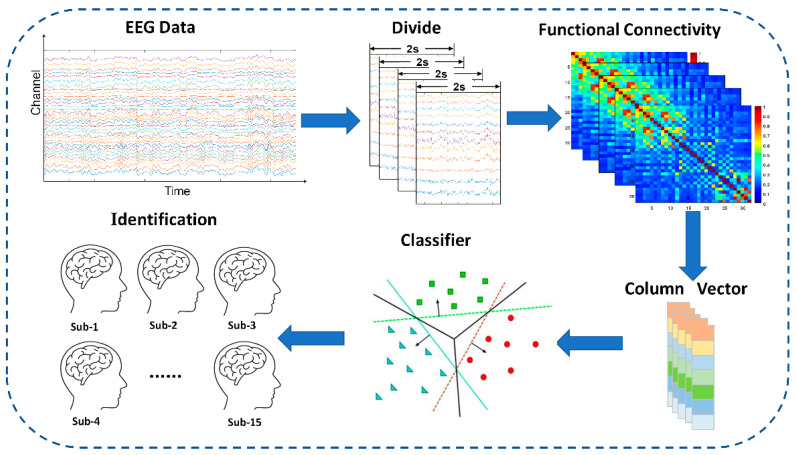
Illustration of identification flow.

**Figure 4 brainsci-11-01266-f004:**
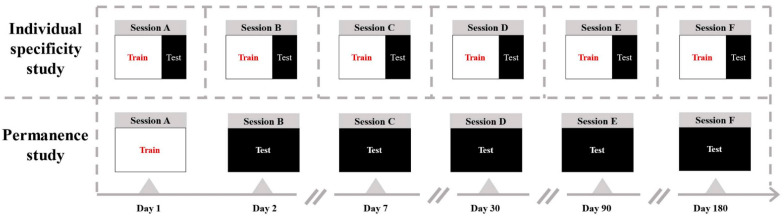
Illustration of individual specificity study in each single session and permanence
study across different sessions. Sessions A to F represent EEG data acquired at 6
different time points over 6 months. The time points were Day 1, Day 2, Day 7, Day 30,
Day 90, and Day 180.

**Figure 5 brainsci-11-01266-f005:**
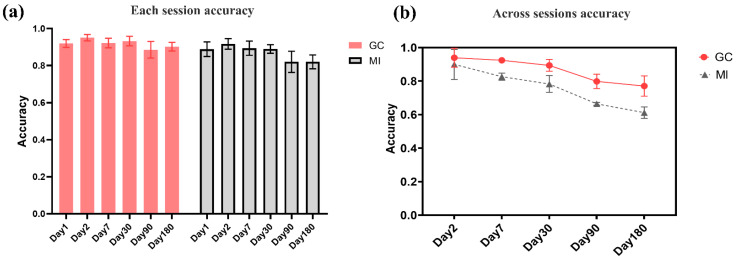
The classification result of intra-sessions (**a**) and inter-sessions
(**b**). The identification accuracy of intra-sessions represented the
specificity of the individual network, and the accuracy of intersessions represented the
permanence of the brain functional network. (**a**) The intra-session
identification accuracy. (**b**) The inter-session identification accuracy.

**Figure 6 brainsci-11-01266-f006:**
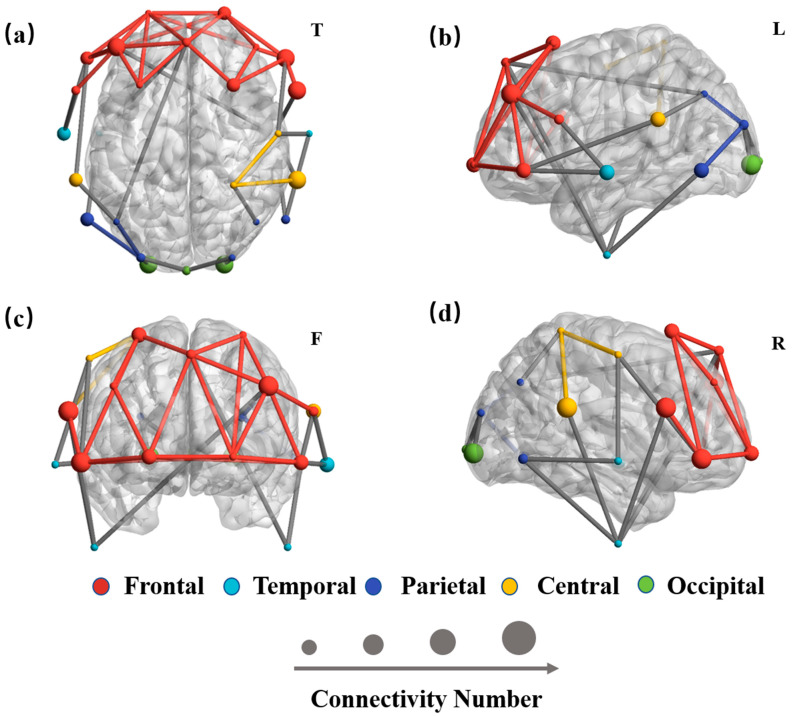
The major connections of individual EEG-FC networks. The size of the spheres represents
the number of major connections to each electrode. Different colored spheres represent
electrodes in different brain regions. The coloured lines indicate connections within
each brain region, and grey lines represent inter-region connections. (**a**)
Top view, (**b**) frontal view, (**c**) left view, and
(**d**) right view.

## Data Availability

The datasets presented in this article are not readily available because the datasets
involve unfinished research projects. If necessary, requests to access the datasets should
contact the corresponding author.

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
