# Peer review of "On the Specificity and Permanence of Electroencephalography Functional Connectivity"

_brainsci, 2021, doi:10.3390/brainsci11101266_

Round 1

Reviewer 1 Report

The article at hand investigates the stability of functional connectivity measures derived from, scalp EEG recordings. In detail, the manuscript describes the investigation of individual stability, i.e. whether connectivity patterns are subject-specific across multiple recordings, and furthermore the permanence of such measures, i.e. whether the derived connectivity patterns remain stable over a time-period of six months. The investigations described are based on two commonly used connectivity estimates, namely Granger Causality derived from autoregressive modelling, as well as the Mutual Information index. With connectivity measures gaining more and more attention in this research area, these kinds of analyses are of increasing importance. Especially the long time period and the resulting high number of repeated EEG recordings can be considered a highlight of the presented research. This is an exciting work, providing evidence of fingerprint-like EEG connectivity patterns that are seemingly stable over quite some time. Multiple interesting possibilities for applications can be thought of in this regard.

However, the description of the chosen methods require some refinement as some aspects are insufficiently explained in order to follow your course of action. Furthermore, there are some minor errors within the methods section that require attention, and in addition some paragraphs are difficult to understand and could be rephrased in order to make them easier to comprehend. Finally, I suggest a proof-reading in order to improve your manuscript in terms of grammar and English.

Please consider the following general suggestions in order to improve your manuscript:

  1. At the end of the introduction section you already describe the main findings (lines 60 ff.). I would suggest to not go into too much detail at this stage of the manuscript. This is more of a summary of results belonging into the conclusion or discussion session. Usually, some hypotheses should be provided here.
  2. Regarding the description of subjects, please include that you mean years of age within the brackets. Second, please refrain from the wording “neurological abnormalities“ - I suggest to use “neurological conditions“. Finally, you should state in this section whether this study was approved of by an ethics committee.
  3. Within section 2.1 you describe the visual capacities of your subjects and then in section 2.2 you state that subjects sat 1m away from a monitor. Did you present anything to the subjects? This is unclear! Please also state whether subjects had their eyes closed during the resting state recording.
  4. On page 2, line 82, please state where on the scalp GND and REF electrodes were positioned (GND and REF are always used as grounding and reference). Also please explain your first preprocessing step. How did you localise electrodes? Did you use individual MRI images or scans? If you simply entered the common 10-20 coordinates, then this is not considered a preprocessing step. Within HERMES this is done automatically, right? So simply state that electrode positions according to the 10-20 system were entered into HERMES for image creation later.
  5. Page 3: In general, there is a rather mixed structure of your reporting. Before showing formulas, you should introduce the actual measure that is derived from the EEG data, i.e. 1. Granger Causality and 2. the mutual information index. After a short introduction with references you can provide the background information regarding the actual calculation of these measurements, and finally you should provide the reader with an idea on how to interpret these, i.e. the range of the values and what they represent in terms of brain connectivity.
  6. Please revise the formulas, as there are some errors and include important information on your models: First, please state that you are calculating the BVAR model, and revise the formulas 1 and 2 to also reflect the model order p. A nice description is provided in: Niso, Guiomar, et al. "HERMES: towards an integrated toolbox to characterize functional and effective brain connectivity." Neuroinformatics 11.4 (2013): 405-434. Second, you should actually state the value of the model order used! This is of great importance for other researchers trying to replicate your results. Third, you should also state whether the model was calculated for all frequencies and in what steps. Finally, formula 3 is wrong (i.e. two summation signs), please revise; a detailed description is provided in: Escudero, Javier, Roberto Hornero, and Daniel Abásolo. "Interpretation of the auto-mutual information rate of decrease in the context of biomedical signal analysis. Application to electroencephalogram recordings." Physiological measurement 30.2 (2009): 187.
  7. Within the methods section after explaining the extracted measures, you should also introduce the machine learning procedure in more detail. What exactly did you classify, against what standard, to what end?!
  8. There is quite some redundancies in the figure descriptions as well as the results section. Both of them should not contain details about what you analysed, but simply what you did find out. In the figure descriptions you need not explain again the validation procedure, but only what is depicted. Furthermore, the descriptions for figure 2 and 3 contain a lot of information on what you calculated, this is not necessary when described thoroughly within the methods section. The same accounts for the results section. You only need to provide the results and not again describe what exactly is depicted in the figures, once already described in its descriptions. Please revise this and delete redundant informations.
  9. Section 3.2 is the same as 3.1, you may have forgotten to enter this correctly. Please revise!
  10. Figure 3, description: I suggest to use more formal terms thanks balls and sticks, i.e. spheres and lines. And also the MRI images’ and angles’ descriptions should adhere to scientific conventions, i.e. axial, coronal and sagittal views. Also delete redundant descriptions in section 3.3.
  11. Within the second paragraph of the discussion section you describe several functions of different cerebral lobes. Please revise this, as all the listed functions are not exclusively represented in specific lobes. Also, different brain regions are not defined according to location. We infer function of specific regions by indirect measurements of functionality recorded at certain sites within the brain!
  12. Within the discussion, several statements should be backed up by citations: (i) lines 219 and 221 on the functions contained within the frontal lobe; (ii) line 223 on consciousness.

Minor/cosmetic suggestion:

  1. page 1, line 10: I suggest to change to: “Electroencephalography-derived functional connectivity…“.
  2. page 1, lines 12f.: I suggest to change to: “EEGFCs remain relatively permanent over time.“, as you did not assess stability across one long recording session.
  3. page 1, line 14: I suggest to delete characteristic here, as it refers to EEG rather than the individual with the sentence being as such.
  4. page 1, line 17: “individually specific“ is misleading, I suggest to use “subject-specific“.
  5. page 1, line 33: “unique and characteristic“.
  6. page 1, line 34: I believe you mean fMRI? MRI itself is not sufficient to extract functional connectivity.
  7. page 1, lines 37ff.: this sounds wrong. Also what do you mean by “normal brain regions“? You could change this to: “Unlike (f)MRI, EEG is a practical and convenient approach to explore the temporal changes in functional brain connectivity, non-invasively recording neuronal activity at the millisecond level.“
  8. page 1, lines 42ff.: please rephrase, e.g.: “… attention regarding the stability of EEG functional connectivity over time and across different cognitive states“.
  9. page 2, line 54: to which study is „their studies“ referring to specifically?
  10. page 2, lines 55ff.: “Therefore, it is necessary to establish a sufficient amount of recordings to firmly demonstrate whether specific functional connectivity patterns remain relatively permanent over time.“
  11. page 2, line 60: permanently exist is misleading here. EEG-FC always does exist but the stability is to be questioned.
  12. page 2, line 73: was quiet.
  13. page 2, lines 75ff: please rephrase this last sentence, e.g.: “Recordings were repeated multiple times for each subject over a period of six months: Day1…, etc.“.
  14. page 2, line 81: electrode T8 is named twice.
  15. page 3, lines 93f.: there is something missing here. I suggest: “… is sufficient for functional connectivity analyses of resting-state EEG data [15].“
  16. page 3, lines 94-97: there is some redundancy here. I suggest to rephrase in order to clarify that you have repeated measure recordings for each subject.
  17. page 3, line 116: you should not use paper in this respect (manuscript or study is better).
  18. page 3, lines 117-119: a verb is missing here.
  19. page 4, line 141: please rephrase. I suggest: “… and 0.82 to 0.92 for the MI network, respectively.“
  20. Section 3.3, page 6: please revise the first sentence as this is common knowledge.
  21. page 6, line 203: delete “the“ before long-term.
  22. page 6, lines 205-209: these statements are rather strong, I suggest to weaken these sentences a littler (e.g. “a potential biological identifier“).
  23. page 7, line 226. What do you mean by functional areas? The entire brain is functional.
  24. page 7: please rephrase the final sentence of the conclusion, also you could give a suggestion for an application and back this up by published research.
  25. page 7, line 248: “… Review Board of the Institute…“.

Author Response

Very grateful to the reviewers' comments, which is helpful for improving our manuscript.The response to the reviewer is attached.

Reviewer 2 Report

The manuscript investigates an interesting and important question concerning the level of permanence and specificity of EEG-based functional connectivity (EEG-FC) within individuals over time. Given the diverse applications of FC-based in across many fields, I believe the topic addressed and the findings reported in the present manuscript will be of interest to a wide range of audiences. I also would like to highlight the well-thought-out experimental and analytical frameworks adopted by the authors. With that said, I do have a few questions that I hope the authors could address.

  1. While the data collection and recording setup is clearly explained, the actual task that the participants were performing while having their EEG concurrently recorded is lacking. From my reading, participants were not performing any active tasks even though there was a monitor screen being set up in the experimental room? Is that correct? In any case, I think it should be clearly explained what exactly the participants were doing during the recording section. This is particularly important for interpreting the present findings as well as for future studies that wish to employ the framework proposed here in other task contexts.
  2. What are the statistical tests used to evaluate the level of significance of the connectivity indexes, Granger causality, and mutual information? Based on the current version of the method section, I believe information regarding the assessment of statistical significance is lacking.
  3. Related to my first comment, can the authors comment on the relationship between the level of FC permanence as a function of task demands. That is, do you think the patterns of observed results are independent of the cognitive tasks individuals engage in? Do you expect to see the frontal lobe regardless of whether individuals are performing a passive or active task and regardless of the type of task demands? Why/why not? I think this is one of the most important points readers would hope to be able to take away from the manuscript, and so I believe some discussion/comment on this question would help put the findings in context for many readers.
  4. The authors mention ‘cognitive-behavioral' differences a few times in the manuscript, but I’m not sure the results really speak to that given the fact that the experimental design doesn’t allow for any behavior (e.g., reaction time, task accuracy) to be measured. And also the cognitive part of this relationship is unclear because participants were not performing an active cognitive task. Overall, I believe the authors should clearly state the limitations of the experimental setup used and explain how the findings can be used to understand what aspects of ‘cognitive-behavioral relationship’.

Author Response

The response to the reviewer is attached.

Round 2

Reviewer 2 Report

I appreciate the thoughtful responses by the authors. All my comments have been carefully addressed.